# Therapeutic Immunomodulation in Gastric Cancer

**DOI:** 10.3390/cancers16030560

**Published:** 2024-01-28

**Authors:** Venu Akkanapally, Xue-Feng Bai, Sujit Basu

**Affiliations:** 1Department of Pathology, The Ohio State University, Columbus, OH 43210, USA; venu.akkanapally@osumc.edu (V.A.); xue-feng.bai@osumc.edu (X.-F.B.); 2Comprehensive Cancer Center, The Ohio State University, Columbus, OH 43210, USA; 3Division of Medical Oncology, Department of Internal Medicine, The Ohio State University, Columbus, OH 43210, USA

**Keywords:** gastric cancer, immunotherapy, immune checkpoint inhibitors, PD-1, PD-L1, CTLA-4, treatment, biomarkers

## Abstract

**Simple Summary:**

Immune checkpoint inhibition is a targeted therapeutic approach in some advanced or metastatic gastric cancer (GC) patients. This review provides elaborated information on various immune therapeutic inhibitors in gastric cancer undergoing clinical trials.

**Abstract:**

Gastric carcinoma, being one of the most prevalent types of solid tumors, has emerged as the third leading cause of death worldwide. The symptoms of gastric cancer (GC) are typically complex, which makes early detection challenging. Immune checkpoint inhibition has become the new standard targeted therapy for advanced or metastatic GC. It is currently being explored in various combinations, both with and without chemotherapy, across multiple therapies in clinical trials. Immunotherapy can stimulate immune responses in GC patients, leading to the destruction of cancer cells. Compared with traditional therapies, immunotherapy has shown strong effectiveness with tolerable toxicity levels. Hence, this innovative approach to the treatment of advanced GC has gained popularity. In this review, we have outlined the recent advancements in immunotherapy for advanced GC, including immune checkpoint inhibitors, cancer vaccines, vascular endothelial growth factor-A inhibitors, and chimeric antigen receptor T-cell therapy. Our current emphasis is on examining the immunotherapies presently employed in clinical settings, addressing the existing challenges associated with these therapeutic approaches, and exploring promising strategies to overcome their limitations.

## 1. Introduction

Gastric cancer (GC), also known as stomach cancer, is the fifth most prevalent cancer worldwide and the third leading cause of cancer-related mortality. Most GC patients are diagnosed at an advanced or metastatic stage, necessitating palliative systemic treatment as the primary therapeutic approach [1,2]. In 2023, the American Cancer Society projected approximately 26,500 new cases of GC in the United States (15,930 in men and 10,570 in women), and the estimated number of deaths from this type of cancer is 11,130 (6690 in men and 4440 in women) [3]. Even with the use of targeted therapy, metastatic disease patients experience a median overall survival (OS) time of just over 1 year [4,5].

The majority of GCs, accounting for approximately 90–95%, are adenocarcinomas originating from gland cells in the innermost mucosal lining of the stomach. There are two primary subtypes of stomach adenocarcinoma. The intestinal type generally has a slightly more favorable prognosis and may exhibit specific gene alterations suitable for targeted drug therapies. In contrast, the diffuse type tends to grow and spread more rapidly, presenting greater treatment challenges [6].

Advances in immuno-oncology and the understanding of the pathophysiology of cancer have significantly transformed cancer treatments. Over the past three decades, three primary treatment approaches—surgery, chemotherapy, and radiation therapy—have dominated the management of upper gastrointestinal malignancies, with surgical resection offering the sole potential for cure. However, this view has shifted with the emergence of immune checkpoint inhibitors (ICIs), challenging the traditional model by delivering unique and sustained periods of improvement in individuals with inoperable conditions. These agents have become the standard of care for various solid tumors, with recent evidence demonstrating their efficacy against gastrointestinal malignancies [7].

Human cancers accumulate numerous genetic and epigenetic changes, leading to the emergence of neoantigens that can be recognized by the immune system. Although preclinical models and patient data revealed the existence of an innate immune response to cancer, its efficacy is limited owing to the development of multiple resistance mechanisms within tumors. These mechanisms include the induction of tolerance, local immune suppression, and systemic disruption of T-cell signaling. Additionally, tumors may employ various strategies to actively escape immune suppression, including intrinsic “immune checkpoint” pathways, which typically curtail immune responses following antigen activation. In response to these findings, extensive efforts have been made to develop immunotherapeutic strategies for cancer, including the development of ICIs, such as anti-programmed death 1 (PD-1), programmed death ligand-1 (PD-L1), and cytotoxic T-lymphocyte-associated antigen-4 (CTLA-4) antibodies for treating advanced cancer patients [8].

Nevertheless, the effectiveness of these traditional therapies has been considerably constrained by multidrug resistance and tumor recurrence. Immunotherapeutic approaches for advanced GC, widely employed among others, include ICIs, adoptive cell therapy, antibodies targeting vascular endothelial growth factor A (VEGFA), cancer vaccines, and chimeric antigen receptor (CAR) T-cell therapy [9,10,11,12]. The role of various immune checkpoint inhibitors in gastric cancer is depicted in Figure 1.

This review explored both established and hypothesized mechanisms of immunotherapy drugs, identified predictive biomarkers for response to ICIs, and examined strategies that have been employed or could be utilized to improve the effectiveness and duration of response to these agents.

## 2. Basic Biology of Immune Checkpoint Inhibitors

### 2.1. Tumor Microenvironment

Tumor tissues and cells exhibit varying degrees of abnormal immune function changes, and an increasing body of evidence indicates the substantial involvement of immunotherapy in shaping the tumor-suppressive immune microenvironment and influencing the progression of GC. The tumor microenvironment (TME) serves as a battleground between tumor cells and antitumor immune cells and possesses distinct characteristics in GC. Within the TME of GC, a diverse array of immune cells can be found, including various cell types such as helper T (Th) cells, regulatory T (Treg) cells, dendritic cells (DCs), tumor-associated macrophages, mesenchymal stem cells, and related inflammatory pathways, all of which have been recognized in cases of GC. Immunotherapy using checkpoint inhibitors is an emerging treatment approach that is rapidly becoming a clinical practice for various cancers [1]. Also, the intratumoral CD4+FOXP3+ T-cells in gastric cancer (GC) engage with nearby immune effector cells, exerting their antitumor effects indirectly rather than through direct contact with tumor cells. Similarly, some studies have demonstrated an elevated number of CD4+FOXP3+ T-cells in close proximity to CD8+ T-cells in GC patients, which was associated with a favorable prognosis. This discovery contributes to advancing our understanding of the function of non-regulatory CD4+FOXP3+ T-cells in cancer. The direct interaction between CD4+FOXP3+ T-cells and CD8+ T-cells revealed a synergistic effect, characterized by a robust IFN-γ response and PDL1 overexpression in GC [13]. Furthermore, research has affirmed significant heterogeneity in the tumor microenvironment across different subtypes. Gastroesophageal adenocarcinomas (GEAs) with MSI-high and EBV positivity exhibited the most pronounced T-cell infiltrates, whereas the GS subgroup demonstrated an abundance of CD4+ T-cells, macrophages, and B-cells. In 50% of GS cases, there was evidence of tertiary lymphoid structures. Conversely, CIN cancers predominantly featured CD8+ T-cells at the invasive margin, with tumor-associated macrophages exhibiting infiltrating capacity [14].

### 2.2. T-Cells

T-lymphocytes play a crucial role in coordinating immune responses. Specifically, targeting and inhibiting these natural regulatory mechanisms or checkpoints could potentially fully activate T-lymphocytes, thereby encouraging more robust antitumor responses. CTLA-4 and PD-1, among the receptors that inhibit the immune response, have undergone extensive development and clinical validation as successful anticancer strategies, utilizing blocking agents [15]. Radiation therapy also stimulates host immunity by inducing immunogenic cell death. This process involves the release of damage-associated molecular patterns, activating DCs. Consequently, DCs can present tumor antigens, initiating the priming of antigen-specific T-cells in a dose-dependent manner [16].

### 2.3. PD-1

PD-1 is a crucial immune checkpoint receptor on activated T-cells, significantly influencing immune suppression. Its primary role unfolds in peripheral tissues, where T-cells interact with PD-1 ligands, such as PD-L1 (B7-H1) and PD-L2 (B7-DC), expressed either by tumor cells, stromal cells, or both. Interference with the interaction between PD-1 and PD-L1 can enhance T-cell response and improve preclinical effectiveness against tumors [8]. PD-1 is primarily found on activated T-cells and plays a negative regulatory role in the immune response by interacting with its ligand PD-L1. PD-L1 is typically present in different immune cell types and in certain cancer cells, allowing these cancers to escape immune detection. Consequently, the expression of PD-L1 in tumors has been explored as a potential prognostic biomarker in various cancer types such as melanoma and non-small cell lung carcinoma [17]. The primary role of cytomembrane-bound PD-L1 is to suppress the antitumor activity of activated T-cells. Consequently, targeting the membranal PD-L1/PD-1 axis is deemed an ideal strategy with significant potential for immunotherapy in lung tumors. Additionally, recent evidence has demonstrated that intracellular PD-L1, functioning as a ribonucleic acid (RNA)-binding protein, can modulate the stability of messenger RNA (mRNA) associated with DNA damage genes. This regulation enhances cellular resistance to DNA damage [18]. A substantial increase in the CD8+ T-cell/eTreg cell ratio was noted in patients undergoing anti-PD-1 therapy, which was linked to a significant increase in the frequency of CD8+ T-cells and a simultaneous reduction in the frequency of eTreg cells within tumor-infiltrating lymphocytes. Notably, these effects were observed even in patients at a progressive disease stage, suggesting that anti-PD-1 therapy may enhance the immune status of the TME, a phenomenon that is distinct from the impact of cytotoxic chemotherapy [19].

### 2.4. PD-L1

PD-L1 belongs to the B7 superfamily of costimulatory molecules found on antigen-presenting cells and acts as an inhibitory factor for T-lymphocytes. Binding to its receptor, PD-1 has been shown to induce T-lymphocyte anergy and/or apoptosis via PD-L1. It has been documented in various malignancies, including colon cancer, ovarian cancer, melanoma, lung carcinoma, breast cancer, non-small cell lung carcinoma, gliomas, squamous cell carcinoma of the head and neck, renal cell carcinoma, and esophageal carcinoma. It induces apoptosis in antigen-specific human T-cell clones and hampers the activation of CD4+ and CD8+ T-cells in in vitro experiments. Numerous studies have shown that antibodies targeting PD-1 or PD-L1 alleviate the inhibitory effects of PD-L1 on cytotoxic T-cells, thereby accelerating the elimination of tumor cells by cytotoxic T-cells [20]. Approximately 50% of GC cells express PD-L1, which is linked to unfavorable prognostic factors such as lymph node metastasis and depth of tumor invasion. Consequently, an immune checkpoint blockade using an anti-PD-L1 antibody can enhance the antitumor immune response in GC or gastroesophageal junction cancer (GEJC) patients [7].

### 2.5. CTLA-4

Initially identified as a receptor expressed on the surface of activated T-lymphocytes with inhibitory functions, CTLA-4 operates differently in resting T-cells. In resting T-cells, CTLA-4 translocates from the Golgi apparatus to the cell surface, where it undergoes immediate endocytosis. CTLA-4 exerts its inhibitory effect by competitively inhibiting the receptor CD28 on the surface of T-cells because both receptors share the same ligands (B7-1 and B7-2). The cytoplasmic portion of CTLA-4 appears to execute an additional inhibitory mechanism by interacting with various signaling molecules. This interaction impedes proximal signaling through the stimulatory receptor CD28 [9,15].

## 3. Immune Checkpoint Inhibitors and Clinical Trials

ICI (ICPI) drugs, encompassing antibodies targeting PD-1, PD-L1, and CTLA-4, have demonstrated the ability to induce sustained complete responses in a subset of patients. Their effectiveness has been observed in both initial and refractory treatment settings for advanced non-small cell lung cancer and melanoma [21]. Various classes of ICIs are available, including monoclonal antibodies directed against PD-1 such as nivolumab and pembrolizumab. Additionally, there are inhibitors for PD-L1, including atezolizumab, avelumab, and durvalumab, as well as CTLA-4 inhibitors such as ipilimumab and tremelimumab [21]. Recent findings support the combination of ICIs with conventional or targeted therapies to enhance the antitumor effects of ICIs and increase the number of patients responsive to these treatments [22]. Innovative strategies are currently being explored to improve the effectiveness of ICIs in the management of solid metastatic malignancies [21]. Although immune checkpoint monotherapy has been proven to be effective in specific situations, combination therapies have shown higher response rates and may be more beneficial for a wider range of patients [23]. Considering the results of the clinical studies that have been summarized in Table 1, there is an unmet need for immuno-oncological treatments that offer a favorable balance between benefits and risks. Dual immunotherapy (nivolumab plus relatlimab) with checkpoint inhibitors has gained attention for prolonging the response duration in responsive patients and improving outcomes in those with disease progression despite treatment [23].

### 3.1. Nivolumab

Nivolumab, a fully human IgG4 monoclonal antibody inhibiting PD-1, in advanced GC or GEJC patients who had received two or more prior chemotherapy regimens showed that nivolumab could potentially be a novel therapeutic option for individuals heavily pretreated for GC or GEJC [7,24]. The ATTRACTION-2 trial is the first randomized controlled study to compare nivolumab to a placebo in Asian patients diagnosed with unresectable or recurrent adenocarcinomas of the stomach or GEJC [24]. All the participants had previously undergone two or more lines of therapy. Median OS was 5.26 months in the nivolumab group and 4.14 months in the placebo group. Notably, the nivolumab group exhibited higher rates of grade 3 and 4 adverse events (10.3%) than did the placebo group (4.3%). In the CheckMate-577 phase III trial, patients with surgically removed (R0) stage II or III esophageal or GEJC who had previously received neoadjuvant chemoradiotherapy and retained residual pathological disease were randomly assigned in a 2:1 ratio to receive either nivolumab or a matching placebo. After a 24-month follow-up in the nivolumab group, the median disease-free survival was 22.4 months, compared to 11.0 months in the placebo group [25]. An ongoing phase II trial, NCT03662659, is actively investigating the potential of nivolumab, either in combination with relatlimab or as monotherapy, in conjunction with chemotherapy for patients diagnosed with GC or gastroesophageal adenocarcinoma [23]. In addition, this study provides comprehensive insights into a study conducted on a cohort of Western patients diagnosed with locally advanced or metastatic esophagogastric cancer. The analyses included safety, efficacy, long-term survival, and biomarker assessment of nivolumab and nivolumab combined with ipilimumab. These findings were derived from a multicenter phase I/II study known as the CheckMate-032 trial. This study showed that nivolumab outperformed the placebo in terms of OS in Asian patients with advanced GC or GEJC [27]. Two phase III trials, RATIONALE-305 and CheckMate-649, confirmed the advantage of incorporating PD-1-targeting drugs with chemotherapy in the initial treatment of advanced GC, GEJC, or esophageal adenocarcinoma [26,31].

### 3.2. Pembrolizumab

Pembrolizumab, the first monoclonal antibody developed to target PD-1, is an IgG4 antibody [10,37]. In the phase II trial KEYNOTE-059 (2017), pembrolizumab monotherapy was administered to advanced GC patients. Notably, the objective response rate (ORR) was 15.5% in the PD-L1-positive patients and 6.4% in the PD-L1-negative patients. The patients were administered intravenous pembrolizumab at a dose of 200 mg every 3 weeks. Monotherapy with pembrolizumab exhibited favorable activity in advanced GC or GEJC patients who had undergone at least two prior lines of treatment, and manageable safety profiles were observed.

Durable responses were observed in patients with both PD-L1-positive and PD-L1-negative tumors, with a 6-month OS of 46.5% (95% CI, 40.2–52.6%) and a 12-month OS of 23.4% (95% CI, 17.6–29.7%) [7,28,38]. In the phase Ib KEYNOTE-012 study, upregulation of PD-L1 expression was observed in certain GC patients. This study assessed the safety and efficacy of the anti-PD-1 antibody pembrolizumab in patients with PD-L1-positive recurrent or metastatic gastric adenocarcinoma. Pembrolizumab demonstrated a manageable toxicity profile and exhibited promising antitumor activity, justifying further investigations in phase II and III trials. The 6-month rate was 66% (95% CI 49–78), and the 12-month rate was 42% (95% CI 25–59) [29,37]. Based on these findings, the FDA approved pembrolizumab as a third-line treatment for PD-L1-positive advanced GC in 2017. The ongoing phase III trial KEYNOTE-062 is investigating the frontline combination of pembrolizumab with cisplatin/5FU in tumors that are PD-L1-positive and human epidermal growth factor receptor (HER2)-negative tumors. Additionally, a randomized, open-label, controlled phase III trial, KEYNOTE-061, compared pembrolizumab with paclitaxel in advanced GC or GEJC patients that progressed after first-line chemotherapy with platinum and fluoropyrimidine [30]. The study found that pembrolizumab did not significantly enhance OS compared to paclitaxel when used as second-line therapy for advanced GC/GEJC in patients with a PD-L1 combined positive score of 1 or higher. Nevertheless, pembrolizumab exhibited a more favorable safety profile than paclitaxel. Multiple additional trials assessing the efficacy of pembrolizumab in GC and gastroesophageal cancer are currently underway [30]. However, a randomized phase III trial, KEYNOTE-061, conducted in advanced GC patients whose disease had progressed after the initial treatment with platinum and fluoropyrimidine doublet therapy, showed that pembrolizumab did not offer a survival advantage over paclitaxel [39]. Pembrolizumab is indicated for the treatment of advanced stomach cancer, typically following other therapies including chemotherapy, under specific conditions, such as a high level of microsatellite instability (MSI-H), defect in a mismatch repair gene, or high tumor mutational burden (TMB-H) [5]. The KEYNOTE-062 trial focuses on advanced, previously untreated GC. It compares the treatment outcomes of pembrolizumab vs. chemotherapy [38]. Tumors harboring a large number of somatic mutations, often caused by mismatch repair defects, may exhibit enhanced susceptibility to immune checkpoint blockade [40]. This suggests that patients with mismatched repair-deficient gastric tumors in these categories may benefit from anti-PD-1 therapy [38,40]. In the United States, pembrolizumab has been approved for the treatment of tumors with mismatch repair deficiencies or MSI [37]. In the pivotal study KEYNOTE-012 conducted in 2014, initial findings were obtained from a cohort of patients diagnosed with advanced GC, all of whom had previously undergone treatment with an anti-PD-1 or anti-PD-L1 monoclonal antibody. These data strongly indicate that the administration of pembrolizumab is not only safe for advanced GC patients but also yields clinically significant antitumor responses, particularly within a population that had received prior treatments [31].

As a result of significant trials, the use of pembrolizumab and nivolumab in combination with chemotherapy has become the standard treatment for metastatic esophagogastric squamous cells and adenocarcinomas. The combination of pembrolizumab with trastuzumab and chemotherapy for HER2-positive esophagogastric cancer is now an established standard of care. These updated treatment guidelines will form the basis for future clinical trials investigating novel therapeutic agents [41]. Nivolumab and CTLA-4 inhibitor ipilimumab operate through distinct yet complementary mechanisms. They contribute to the restoration of antitumor T-cell function and induction of de novo antitumor T-cell responses, respectively. The OS in patients with a PD-L1 combined positive score ≥ 5 for nivolumab plus ipilimumab, compared to chemotherapy alone, did not reach the prespecified boundary for significance. No new safety signals were observed. These findings support the continued use of nivolumab plus chemotherapy as a standard first-line treatment for advanced gastroesophageal adenocarcinoma [42,43]. Perioperative immunotherapies useful in certain circumstances like MSI-H/dMMR tumors includes (1) nivolumab and ipilimumab followed by nivolumab, (2) pembrolizumab, (3) tremelimumab and durvalumab for neoadjuvant therapy only according to NCCN guidelines [44]. INFINITY is an ongoing phase II clinical trial with the primary objective of evaluating the efficacy of combining immunotherapies tremelimumab and durvalumab in the neoadjuvant or definitive treatment of resectable MSI-H GC/GEJC [38]. Moreover, PD-L1 blocking therapy is employed in conjunction with various widely used tumor therapies, including chemotherapy, radiotherapy (RT), photodynamic therapy (PDT), photothermal therapy (PTT), adoptive cell therapy (ACT), oncolytic viral therapy, and bacterial therapy. This is because these therapies can, to some extent, elevate the expression of PD-L1. Consequently, PD-L1 blocking therapy is considered a crucial and effective strategy for immune checkpoint blockade, reactivating cytotoxic T-cells and thereby reversing tumor immune suppression. Also, the dual-immune regulation strategy targeting tumor-responsive PD-L1 and Cox-2 proved to be more effective compared to strategies focused solely on PD-L1 or Cox-2 in inhibiting tumor growth, preventing metastasis, and avoiding relapse. This was observed even in the case of low immunogenic tumors, such as 4T1 breast cancer [18].

### 3.3. Toripalimab

In a study investigating the safety and effectiveness of toripalimab, a humanized PD-1 antibody, in advanced GC patients, the research also aimed to assess potential predictive survival advantages associated with TMB and PD-L1. In cohort 1, advanced GC patients received toripalimab as a monotherapy, and in cohort 2, advance GC patients received a combination of toripalimab and XELOX (oxaliplatin and capecitabine) as a 3-week treatment cycle. In addition, toripalimab maintains a manageable safety profile and exhibits promising antitumor activity in advanced GC. As a monotherapy, toripalimab demonstrates an objective response rate (ORR) comparable to that of nivolumab and pembrolizumab in patients who are not selected based on their PD-L1 status and have undergone extensive prior treatment [39].

Secondly, PD-L1 ICIs revealed an increase in PD-L1 expression in 50% of GC patients, and this increase was associated with poorer survival outcomes [7]. The effect of PD-L1 on tumor-infiltrating immune cells varies in response to different treatment regimens for various malignancies. Furthermore, even at different stages of the same tumor type, the pattern of changes may not be consistent. The findings of their study make a significant contribution as they are the first to reveal variations in the expression of multiple checkpoint molecules, extending beyond PD-L1. Additionally, the study includes markers of tumor-infiltrating immune cells in the context of neoadjuvant chemotherapy for locally advanced GC. An important highlight of their research was the identification of a robust positive correlation between the changes in the expression of another checkpoint molecule, TIM3, and those of PD-1 and PD-L1 [4]. PD-L1 upregulation has been observed in certain GC patients. The goal of the phase Ib KEYNOTE-012 study was to assess the safety and effectiveness of pembrolizumab, an anti-PD-L1 antibody, in individuals with recurrent or metastatic adenocarcinoma of the stomach or GEJ who tested positive for PD-L1 [29]. Among patients with recurrent or metastatic PD-L1-positive GC, pembrolizumab exhibited a manageable toxicity profile and promising antitumor activity, thereby justifying further investigation in phase II and III trials [29]. The upregulated expression of PD-L1 and APE1 has been linked to the development of GC and poor prognosis. This study revealed that elevated levels of PD-L1 and APE1 are risk factors for GC and represent novel biomarkers for predicting prognosis [20].

### 3.4. Durvalumab

Durvalumab, a monoclonal antibody targeting PD-L1, inhibits the interaction between PD-L1, PD-1, and the CD80 receptors on T-cells. This effectively eliminates the inhibitory PD-1/PD-L1 pathway, which is commonly activated in the TME. This unblocking process reestablishes an efficient antitumor T-cell response [22]. Early-phase clinical studies have provided strong evidence for advancing the clinical development of the ICI durvalumab as an anti-PD-L1 antibody for GC or GEJC patients. A growing body of evidence suggests that combining ICIs with 5-fluorouracil, leucovorin, oxaliplatin, and docetaxel (FLOT) chemotherapy can enhance clinical outcomes in advanced or metastatic cancer patients [34]. A phase Ib/II randomized, multicenter, open-label study examined the combination of durvalumab and tremelimumab or their efficacy as monotherapy in chemotherapy-refractory GC or GEJC patients. The combination of ramucirumab and durvalumab showed manageable safety and antitumor activity across all cohorts, with particularly notable effects on patients exhibiting high PD-L1 expression [22]. Durvalumab monotherapy is the first-line treatment for unresectable stage III non-small cell lung cancer following platinum-based chemoradiotherapy (specific to PD-L1 positive patients in the European Union). Furthermore, durvalumab, either alone or in combination with tremelimumab, a CTLA-4 inhibitor, has shown preliminary clinical activity against GC/GEJ adenocarcinoma and hepatocellular carcinoma [22].

A phase Ib/II randomized, multicenter, open-label study examined the combination of durvalumab and tremelimumab or their efficacy as monotherapy in chemotherapy-refractory GC or gastroesophageal junction GEJC patients. The OS rate at 12 months in the combination therapy arm was 37.0% (95% CI, 19.6–54.6%), whereas it was 4.6% (95% CI, 0.3–19.0%) for durvalumab monotherapy and 22.9% (95% CI, 3.5–52.4%) for tremelimumab monotherapy [33].

One study investigated the use of avelumab, an anti-PD-L1 therapy, as a maintenance treatment following first-line induction chemotherapy for GC or GEJC patients. The results of the JAVELIN Gastric-100 study indicated that avelumab maintenance did not significantly improve OS compared with the continuation of chemotherapy, and this was observed in advanced GC or GEJC patients, both in the overall study population and in a predefined PD-L1-positive subgroup [45]. Currently under investigation for various disease conditions, tislelizumab, an experimental monoclonal antibody targeting PD-1, is being studied both as a standalone treatment and in combination with other therapies [31,46].

### 3.5. Atezolizumab

In the randomized phase II DANTE trial conducted by the Arbeitsgemeinschaft Internistische Onkologie, the experimental anti-PD-L1 antibody atezolizumab was evaluated in patients with resectable localized EGC. Patients were allocated randomly in a 1:1 ratio to either the experimental arm, which included atezolizumab in combination with chemotherapy (FLOT), or the standard arm, who received mono-chemotherapy. The main outcomes under consideration were progression-free survival and disease-free survival. The preliminary safety analysis indicated that the perioperative use of atezolizumab with FLOT is both feasible and safe; we are awaiting the future efficacy results [47]. In patients diagnosed with metastatic renal cell carcinoma, the amalgamation of atezolizumab, an anti-PD-L1 agent, with bevacizumab, an anti-VEGFA drug, showed improved clinical responses. Additionally, there was a substantial elevation in intratumoral CD8+ T-cell and chemokine levels [22]. Data, also from a German phase II-III study, show that the standard of care for patients capable of tolerating a triple cytotoxic drug regimen should involve the perioperative use of FLOT, with four cycles administered both pre- and post-operatively [48].

Numerous studies have been conducted to assess the safety and effectiveness of combining immunotherapies such as anti-TIM3 and anti-PD-1, either alone or in combination with chemotherapy, and explored across diverse tumor types, including GC. The outcomes of these studies were anticipated (ClinicalTrials.gov identifiers: NCT03469557, NCT02608268, NCT02817633, NCT03448835, and NCT03399071). TIM3, identified as an immune checkpoint receptor, is located on interferon (IFN)-γ-secreting T helper (Th)-1 cells, natural killer (NK) cells, and CD8+ cells. Its role involves hindering the activation of T and NK cells by interacting with its ligand, galectin 9. This upregulation of TIM3 expression may be linked to adaptive resistance to PD-1 blockade. One study highlighted the upregulation of TIM3 expression and its positive association with PD-1 and PD-L1 expression following neoadjuvant chemotherapy. The strong positive correlations observed between TIM3, PD-1, and PD-L1 expression in this study suggest that dual-target immunotherapy involving PD-(L)1 and TIM3 could be a valuable option for GC patients [4].

Moreover, innovative treatment approaches, such as utilizing anti-carcinoembryonic antigen (CEA) bispecific T-cells, have generated impressive responses in colorectal cancer, which is characterized by low MSI. These advancements may pave the way for novel strategies to manage upper GI malignancies, especially HER2-positive gastroesophageal cancers for which specific biomarkers have been identified [7].

CTLA-4 plays a pivotal role in human immunity. It shares homology with CD28 but interacts with B7-1/B7-2 with higher affinity [49]. Consequently, CTLA-4 can modulate or impede CD28 signaling. Clinical trials have investigated CTLA-4 inhibitors, including tremelimumab and ipilimumab, in advanced GC. A phase II trial assessing ipilimumab in advanced GC patients was prematurely terminated as it did not demonstrate a substantial enhancement in survival rates compared with first-line targeted agents [32]. In a clinical study of tremelimumab that included 12 patients diagnosed with inoperable advanced GC, a moderate response rate was observed, particularly when the drug was used in combination with other anticancer agents. Notably, combination therapies concurrently targeting CTLA-4 and PD-1 have shown augmented antitumor immune responses [33]. Combination therapy with ipilimumab and nivolumab has gained approval for the treatment of advanced GC. Nevertheless, the effectiveness of CTLA-4 inhibitors as standalone treatments for advanced GC requires further investigation [9]. Tremelimumab, formerly known as ticilimumab or CP-675,206, is a monoclonal antibody that targets CTLA-4. It has not received FDA approval but has obtained orphan drug status for mesothelioma. Researchers are currently exploring its potential application in GEJ and gastric adenocarcinoma, both as a standalone treatment and in combination with durvalumab [46].

### 3.6. Toxicity Profile of ICIs

Based on the findings outlined in above clinical studies, it can be deduced that the immunotherapy strategy in gastric cancer (GC) has achieved only moderate success. These agents were associated with treatment-related toxicities, frequently of grade 3 or higher, leading to treatment-related fatalities. The primary reported adverse events encompassed fatigue, diarrhea, pruritus, rashes, gastrointestinal bleeding, pneumonia, urinary tract infections, gastrointestinal toxicity, loss of appetite, arthralgia, hypothyroidism, ALT increase, and colitis. And toxicity profile of various clinical trials is summarized in Table 2.

### 3.7. Anti-Lymphocyte Activation Gene 3 (LAG-3)

Relatlimab, a groundbreaking first-in-class blocking antibody, was designed to target LAG-3, a protein expressed on lymphocyte surfaces that plays a role in restraining T-cell proliferation and encouraging T-cell exhaustion within the tumor’s immune microenvironment [23].

### 3.8. Radiotherapy Subsequent to Anti-PD-1 Therapy

This study explored the efficacy of radiotherapy in metastatic gastric cancer (mGC) patients who had previously undergone anti-PD-1 treatment. According to computed tomography (CT) findings, 28% of mGC patients who were treated before anti-PD-1 therapy and then subjected to radiotherapy demonstrated a favorable tumor response. Conversely, patients who did not undergo the initial anti-PD-1 treatment did not respond to radiotherapy. These findings suggest that anti-PD-1 therapy enhances the sensitivity of mGC to radiotherapy. This study highlights a significant response to radiotherapy observed in mGC patients following prior anti-PD-1 therapy [19].

Furthermore, multiple preclinical studies have reported that irradiation increases the expression of PD-L1, an important immune checkpoint molecule. The expression of PD-L1, regulated by the IFN-dependent pathway, was upregulated after irradiation. While both type I and type II IFNs can enhance PD-L1 expression, IFNγ, a member of the type II IFN family, exhibits stronger and more persistent effects via the JAK–STAT–IRF pathway [16].

### 3.9. Regulatory T-Cells (Tregs)

Tregs promote tumor growth by modulating the antitumor immune response, primarily by inhibiting T-cell-mediated tumor cell destruction. This inhibitory effect is believed to be dependent on the actions of IL-10 and/or TGF-b. Tregs are typically identified by the expression of CD4, elevated levels of CD25 (CD25 high), and the presence of the transcription factor Forkhead box P3 (FOXP3), which is associated with suppressive activity. FOXP3 serves as a master regulator of Tregs, forms complexes with various proteins, and undergoes diverse post-translational modifications (PTMs), including acetylation, phosphorylation, ubiquitination, and methylation. These PTMs influence the stability of FOXP3 and its ability to regulate gene expression, ultimately affecting Treg activity [2,50].

### 3.10. CAR T-Cells

Antibodies and CAR T-cells are increasingly used in cancer immunotherapy [51]. CAR T-cell therapy involves engineering T-cells to express synthetic receptors designed to recognize and target specific cancer antigens. This modification activates T-cells, enabling them to activate the immune system to identify and eliminate tumor cells. Crucial roles in the diagnosis and functionality of GC are played by biomarkers like claudin 18.2 (CLDN 18.2), HER2, mucin 1, natural killer receptor group 2, epithelial cell adhesion molecule, mesothelin, and CEA [42,52]. Research findings indicate that CAR-T cells can effectively target these biomarkers as a treatment option for advanced GC. HER2 is an overexpressed surface antigen found in GC cells. HER2-positive GC typically demonstrates resistance to multiple drugs, hindering the effectiveness of traditional treatments. The development of drug resistance poses a substantial challenge in the treatment of advanced GC [10,42]. Importantly, CAR T-cell therapy has emerged as a promising approach to combat multiple drug resistance observed in advanced GC patients. Research on HER2CAR T therapy has shown a strong affinity for treating advanced GC. Clinical trials involving CLDN18.2 CAR T-cells, specifically designed for CLDN18.2-positive patients, have shown remarkable antitumor efficacy in tumor models. CA 72-4, a surface glycoprotein that is highly prevalent in advanced GC, has become a promising target for CAR T-cell therapy, demonstrating significant potential in eliminating tumors. Clinical studies have revealed that combining CAR T-cell therapy with other treatment modalities leads to enhanced antitumor effects [9]. Furthermore, CAR T-cell therapies directed against B7-H3 and CDH17 have shown significant promise in cancer treatment. Clinical investigations have revealed that B7-H3 is overexpressed in tumor tissues of advanced GC patients and is closely associated with disease progression. In advanced GC patients, assessments of B7-H3-specific CAR T-cells have demonstrated significant antitumor potential and cytotoxicity against gastric tumor cells. CDH17, a biomarker of gastrointestinal adenocarcinomas, plays a crucial role in calcium-dependent adhesion switching and Wnt signaling [53]. Recent advancements in CAR T-cell therapies targeting CDH17 have illuminated this novel immunotherapeutic approach as a potentially safe and effective treatment option for advanced GC. Studies in preclinical mouse models with gastrointestinal carcinoma xenografts have shown the strong efficacy of CDH17CAR T therapy against advanced GC, while sparing normal gastrointestinal epithelial cells from notable toxicity [9].

### 3.11. Tumor Antigen Vaccines

Tumor antigen vaccines are produced using cancer cells, fragments of cancer cells, or purified tumor antigens extracted from tumor cells. This vaccine aims to activate the immune system, prompting the identification and elimination of cancer cells. Numerous tumor antigens have been studied to assess their effectiveness as antitumor agents. In previous studies on GC, tumor peptide vaccines such as G17DT, vascular endothelial growth factor receptor (VEGFR), and OTSGC-A24 were assessed for antitumor activity. G17DT, a vaccine engineered to counteract gastrin-17, a hormone essential for the growth of multiple gastrointestinal tract cancers, is one such vaccine. Studies have also shown that G17DT is well tolerated and effective in treating advanced cancer patients [10].

### 3.12. Combination of Anti-VEGF Drugs with ICIs

Recent research indicates that antiangiogenic therapies may not only exert direct antiangiogenic effects but also possess immunomodulatory properties. Notably, a recent preclinical study demonstrated the upregulation of PD-L1 by IFN-γ-expressing T-cells in mouse models of refractory pancreatic, breast, and brain tumors. These findings provided a compelling basis for considering their combination with ICIs (ICPI) [21]. Increasing evidence indicates the potential of enhancing ICI antitumor activity and broadening the spectrum of patients who respond to ICIs by combining them with conventional or targeted therapies. Blocking the VEGF pathway has the potential to transform the immunosuppressive TME into an immune-supportive or T-cell-inflamed environment, which could enhance the effectiveness of ICIs and broaden the range of patients who can benefit from them. In advanced GC/GEJC patients, the targeting of VEGFR2 with ramucirumab (RAM) led to increased infiltration of CD8+ T-cell infiltration and PD-L1 expression within the TME [22]. The REGARD trial is one of the largest phase III trials for the second-line treatment of GC/GJEC. This study compared ramucirumab monotherapy with a placebo in gastric/GEJ adenocarcinoma patients. In the ramucirumab group, the median OS was 5.2 months (IQR 2.3–9.9), whereas in the placebo group, it was 3.8 months (1.7–7.1) (hazard ratio (HR) 0.776, 95% CI 0.603–0.998; *p* = 0.047). This study reported higher rates of hypertension as an adverse effect in the ramucirumab group than in the placebo group [35]. The RAINBOW study demonstrated that the combination of ramucirumab with paclitaxel increased OS of patients previously treated for advanced GC compared to the placebo plus paclitaxel group. The group receiving ramucirumab plus paclitaxel demonstrated a significantly longer OS than the placebo plus paclitaxel group, with a median of 9.6 months (95% CI 8.5–10.8) versus 7.4 months (95% CI 6.3–8.4) [36].

Ramucirumab plus durvalumab demonstrated manageable safety, and the combination exhibited antitumor activity across all cohorts, with notable effectiveness in patients showing high PD-L1 expression [22].

Moreover, when used as a first-line treatment, the combination of bevacizumab, a monoclonal antibody directed against VEGF-A, in combination with chemotherapy has been associated with significantly improved rates of ORR and progression-free survival (PFS) in metastatic GC patients. Although the improvement in OS was not statistically significant, these findings highlight the importance of VEGFR-2 signaling as a valuable therapeutic target for advanced gastric and gastroesophageal junction adenocarcinomas [36].

Apatinib, or rivoceranib, is a selective tyrosine kinase inhibitor that targets VEGF receptor 2. Despite being investigated for its efficacy in metastatic GC, outcomes have varied. Despite the lack of evident benefits, as suggested by the undisclosed results unveiled in ESMO 2019, apatinib has not gained acceptance as a conventional treatment for metastatic GC except in China [54].

Trastuzumab, a monoclonal antibody targeting the HER2, also known as ERBB2, has been investigated in combination with first-line chemotherapy for the treatment of advanced HER2-positive gastric or gastroesophageal junction cancers. These findings suggest that a combination of trastuzumab and chemotherapy is a new standard treatment option for HER2-positive advanced GC or GEJC patients [5]. In addition, one study showed that HER2 gene amplification, as determined by the HER2/CEP17 ratio and HER2 gene copy number, could significantly predict improved OS and treatment response in advanced GC patients undergoing trastuzumab-based chemotherapy [55].

## 4. Immune Checkpoint Based on Molecular Classification of Gastric Cancer

GC represents a significant contributor to cancer-related mortality, yet its complex molecular and clinical characteristics have posed challenges owing to histological and etiological diversity. A comprehensive molecular analysis was performed on 295 primary gastric adenocarcinomas within the Cancer Genome Atlas project to address this complexity. This endeavor led to the proposal of a molecular classification system that divides GC into four distinct subtypes:

**1. Epstein–Barr Virus-Positive (EBV+) Tumors:** These tumors feature recurrent PIK3CA mutations, extensive DNA hypermethylation, and amplification of genes such as JAK2, CD274 (also known as PD-L1), and PDCD1LG2 (also known as PD-L2). Notably, EBV has been detected in the malignant epithelial cells in approximately 9% of GC cases.

**2. Microsatellite Unstable Tumors:** This subtype exhibits elevated mutation rates, including mutations affecting genes encoding oncogenic signaling proteins that are targeted for therapy.

**3. Genomically Stable Tumors:** Enriched by diffuse histological variants, these tumors often harbor mutations in RHOA or fusions involving RHO family GTPase-activating proteins.

**4. Tumors with Chromosomal Instability:** This subtype demonstrates marked aneuploidy and focal amplification of receptor tyrosine kinases.

The identification of these subtypes offers valuable insights into patient stratification and paves the way for targeted therapy trials. Moreover, stomach cancers associated with EBV tend to exhibit slower growth and a reduced propensity for metastasis [15,56]. Considering the emerging data on the significance of MSI as a predictive biomarker for the response to ICIs, exploratory analyses have shown promising responses in patients with both MSI-high and non-MSI-high tumors. While the ORR was numerically higher in the MSI-high subgroup, it is essential to note that these findings were hypothesized to be generated owing to the relatively small sample size. Further studies with larger patient cohorts are required to validate these findings [27]. In 2021, a meta-analysis incorporating four randomized clinical trials (KEYNOTE-062, CheckMate-649, JAVELIN Gastric-100, and KEYNOTE-061) revealed a significant enhancement in OS within the MSI-high cancer subgroup compared to the microsatellite stable subgroup [54].

Gastric cancer with lymphoid stroma (GCLS) is a rare type of gastric cancer characterized by abundant lymphocytic infiltration of the stroma. It is an Epstein–Barr virus-associated gastric cancer with a better prognosis than typical gastric cancer but with similar symptoms. Inspection of the gastrectomy specimen (surgical specimen) and in situ hybridization revealed the presence of the Epstein–Barr encoding region. GCLS diagnosis is based on pathological, histological, and immunohistochemical examination and there are no standardized guidelines for treatment. The patient recovered well after immunotherapy [57].

Furthermore, elevated TMB correlates with improved clinical response. Patients with TMB-high tumors demonstrated significantly superior responses compared to those with TMB-low tumors. Notably, the TMB-high group exhibited a substantial survival advantage in OS, amounting to approximately 4.0 months [39].

## 5. Conclusions and Future Directions

In recent years, the survival outlook for GC patients has improved not only through chemotherapy but also via targeted therapies [39]. Recent advancements in understanding the biology of GC have revealed elevated expression rates of immune checkpoints, along with their specific variations, in GC tumor tissues compared to normal controls. These studies have highlighted the potential significance of immune checkpoint aberrations in shaping the microenvironment and contributing to the development and progression of GC. Following the evidence of the efficacy and safety of immunotherapy for GC, the next pivotal challenge is to determine the optimal treatment approach. Exploration of targeted immune checkpoints has evolved from single-drug treatments to combination therapies. Strategies combining immunotherapy with chemotherapy have been implemented in various clinical settings. Predominant results from studies on GC have indicated that, compared with chemotherapy alone, the combination of immunotherapy and chemotherapy enhances treatment efficacy to varying extents. Nevertheless, the optimal practice for integrating chemotherapy with immunotherapy requires further investigation because of the associated side effects of chemotherapy. Additional approaches to enhance the utilization of ICIs include dual checkpoint inhibition, integration of chemotherapy with checkpoint inhibitors, and a combination of checkpoint inhibition with other investigational immunotherapeutic options. It is crucial to conduct additional fundamental and clinical investigations to enhance and optimize this approach for GC and other gastrointestinal malignancies.

The progress in clinical research over the past two decades, coupled with the approval of various targeted and immunomodulatory agents, alongside chemotherapeutic agents has notably enhanced the treatment paradigm for advanced gastric cancer. Despite significant benefits demonstrated by these agents for a subset of patients, the duration of these benefits is limited. Managing and treating the disease becomes a significant challenge in its advanced stage. Although various growth factors regulate distinct immune checkpoint inhibitors, most preclinical studies and clinical trials have primarily focused on anti-PD-L1 or anti-PD-1, revealing modest clinical responses and adverse events in patients. Moreover, conducting in-depth studies to identify more effective drug combinations, minimize toxicity, determine the optimal dosage for each drug in a combination, and monitor pharmacodynamic endpoints is essential. Importantly, a comprehensive understanding of the immune checkpoint process and the signaling pathways that regulate it is crucial to design novel targeted therapies for gastric cancer.

While immunotherapy holds promise for advanced gastric cancer, challenges like modest clinical efficacy and immune evasion hinder its widespread application. Addressing these challenges will entail combining CAR T therapy with immune checkpoint inhibitors (ICIs) and employing immune modulators to prevent immune suppression. The development of innovative immunotherapies is expected to provide insights into the treatment of advanced gastric cancer.

Overall, immunotherapy has become a new and innovative treatment approach for some individuals with upper gastrointestinal malignancies. Continuing research efforts aim to uncover additional biomarkers predicting responses to immunotherapy, going beyond PD-L1 expression and tumor mutation burden. The application of mass spectrometry and advanced bioinformatic algorithms holds promise in identifying neoepitopes that are likely to provoke an immune response. These tools are anticipated to play a crucial role in enriching clinical trials with patient subsets more likely to derive benefits from immunotherapy in the future.

## Figures and Tables

**Figure 1 cancers-16-00560-f001:**
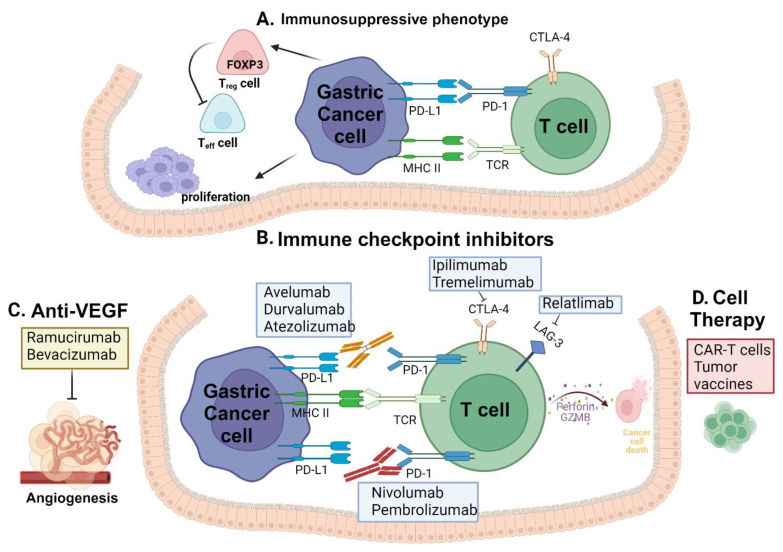
Immune checkpoint inhibitors in gastric cancer: (**A**) The interaction between PD-L1 on tumor cells and PD-1 on T-cells that fight against the tumor suppresses T-cell activation. (**B**) Inhibiting the interaction between PD-L1 and PD-1 is achieved by blocking PD-1 with an anti-PD-1 antibody and similarly PD-L1 with an anti-PD-L1 antibody. This blockade prevents checkpoint inhibition, enabling antitumor T-cell function. Blocking of CTLA-4 and LAG3 allows for the T-cell to interact with tumor cells. (**C**) Anti-VEGF drugs in combination with anti-PD-1 and anti-PD-L1 drugs inhibiting the angiogenesis pathway. (**D**) Cell therapy includes engineered T-cells and tumor antigen vaccines for cancer immunotherapy. Figure was created using BioRender.com.

**Table 1 cancers-16-00560-t001:** Clinical trials of immune checkpoint inhibitors.

S. No	Clinical Trial	Drug	Phase	Disease	Target Receptor	Overall Survival	Ref.
1	ATTRACTION-2	Nivolumab vs. placebo	Phase III trial	Advanced GC/GEJC	PD-1	Median OS (95% CI) nivolumab group was 5.26 months and placebo group was 4.14 months HR 0.63 (95% CI 0.51–0.78); *p* < 0.0001	[24]
2	CheckMate-577	Nivolumab as adjuvant therapy	Phase III trial	Esophageal or GEJC	PD-1	Nivolumab group OS was 22.4 months (95% CI, 16.6 to 34.0) and placebo group OS was 11.0 months (95% CI, 8.3 to 14.3) HR 0.69 (96.4% CI, 0.56–0.86); *p* < 0.001	[25]
3	CheckMate-649	Nivolumab/ipilimumab vs. nivolumab vs. chemotherapy	Phase III	Advanced GC/GEJ	PD-1	Nivolumab plus chemotherapy OS 13.1 and chemotherapy alone OS 11.1. HR 0.71(98.4% CI 0.59–0.86); *p* < 0.0001	[26]
4	CheckMate-032	Nivolumab and nivolumab plus ipilimumab	Phase I/II trial	Metastatic GEJC	PD-L1+ and PD-L1 and MSI	Median OS (95% CI) NIVO3 was 6.2 (3.4 to 12.4) NIVO1 + IPI3 was 6.9 (3.7 to 11.5) NIVO3 + IPI1 was 4.8 (3.0 to 8.4)	[27]
5	KEYNOTE-059	Pembrolizumab	Phase II	GC/GEJC	PD-L1	Median OS was 16.3 (1.6+ to 17.3+) months for PD-L1-positive and 6.9 (2.4 to 7.0+) months for PD-L1-negative	[28]
6	KEYNOTE-012	Pembrolizumab	Phase Ib trial	Advanced GC	PD-1	Median OS was 11.4 months (95% CI 5.7 not reached).	[29]
7	KEYNOTE-61	Pembrolizumab versus paclitaxel	Phase III	Advanced GC/GEJC	PD-1	Pembrolizumab median OS was 9.1 months (95% CI 6.2–10.7) for and paclitaxel median OS 8.3 months (95% CI 7.6–9.0)	[30]
8	RATIONALE-305	Tislelizumab plus ICC and placebo plus ICC	Phase III	Advanced GC/GEJC	PD-1	TIS + ICC median OS is 17.2 (95% CI: 0.55–0.83) vs. P + ICC median OS is 12.6 months (95% CI: 0.59–0.94)	[31]
9	NCT01585987	Ipilimumab monotherapy vs. best supportive care (BSC)	Phase II	Advanced GC/GEJC	CTLA-4	Ipilimumab monotherapy median OS was 12.7 months (95% CI, 10.5–18.9) and BSC group median OS was 12.1 months (95% CI, 9.3–not estimable), study ceased	[32]
10		Durvalumab alone (B) and tremelimumab alone (C) or in combination(A)	Phase Ib/II	Advanced GC/GEJC	PD-L1 and CTLA-4	Median OS for combination (A) was 9.2 months (95% CI, 5.4–12.6 months), alone (B) median OS was 3.4 months (95% CI, 1.7–4.4 months), alone (C) median OS was 7.7 months (95% CI, 2.1–13.7 months)	[33]
11	MATTERHORN	Durvalumab plus FLOT	Phase III	GC/GEJC	PD-L1	Ongoing (until February 2025)	[34]
12		Ramucirumab and durvalumab	Phase Ia/b	Advanced GC/GEJC	PD-L1	Patients showed median OS was 12.4 (95% CI, 5.5–16.9) and patients with high PD-L1 expression median OS was 14.8 months	[22]
13	REGARD	Ramucirumab monotherapy vs. placebo	Phase III	Advanced GC/GEJC	VEGF and VEGFR2	Ramucirumab median OS was 5.2 months (2.3–9.9) and median OS for placebo was 3·8 months (1.7–7.1)	[35]
14	RAINBOW	Ramucirumab plus paclitaxel vs. placebo plus paclitaxel	Phase III	Advanced GC/GEJC	VEGFR2	Median OS for ramucirumab plus paclitaxel was 9.6 months (95% CI 8.5–10.8) and for placebo plus paclitaxel was 7.4 months (95% CI 6.3–8.4)	[36]
15	ToGA (trastuzumab for gastric cancer)	Trastuzumab plus chemotherapy vs. chemotherapy alone	Phase III	Advanced GC/GEJC	HER-2-positive	Median OS for trastuzumab plus chemotherapy 13·8 months (95% CI 12–16) compared with chemotherapy alone 11·1 months (10–13) (hazard ratio 0.74; 95% CI 0∙60–0∙9)	[5]

**Table 2 cancers-16-00560-t002:** Toxicity profile of various clinical trials.

S.No	Drug and Mechanism of Action	Adverse Events	Treatment-Related Effects and Deaths	Ref No
1	Nivolumab Placebo	Fatigue Diarrhea Pruritus Rash	34% in nivolumab group showed adverse effects. 32% in placebo group showed adverse effects. Serious adverse events occurred in 30%.	[25]
2	Nivolumab and placebo	Pruritus Diarrhea Rash Fatigue Colitis Pyrexia Pneumonia Urinary tract infection	43% in nivolumab group showed adverse effects. 27% in placebo group adverse effects. Deaths 2% death in nivolumab group. 1% death in placebo group.	[24]
3	Nivolumab + chemotherapy Chemotherapy alone	Nausea Diarrhea Peripheral neuropathy Pneumonia GI toxicity GI bleeding Diarrhea Asthenia Loss of appetite Pneumonitis	59% in nivolumab + chemotherapy group showed adverse effects. 44% in chemotherapy alone group show adverse effects. Deaths 2% deaths in nivolumab group. 1% deaths in chemotherapy alone group.	[26]
4	Nivo alone Nivo (1 mg/kg) + Ipilimumab (3 mg/kg) Nivo (3 mg/kg) + ipilimumab (1 mg/kg)	Decreased appetite. Diarrhea Fatigue Pruritus Rash	17% in nivo alone group showed adverse effects. 47% in nivo (1 mg/kg) + ipilimumab(3 mg/kg) group showed adverse effects. 27% in nivo (3 mg/kg) + ipilimumab (1 mg/kg) group showed adverse effects.	[27]
5	Pembrolizumab	Fatigue Pruritus Rash Hypothyroidism Anemia Nausea Diarrhea Arthralgia	17.8% of pembrolizumab group showed adverse effects.	[28]
6	Pembrolizumab	Appetite Hypothyroidism Pruritus Arthralgia	67% patients have treatment related adverse effects. 13% showed grade 3–4 adverse effects.	[29]
7	Pembrolizumab Paclitaxel	Anemia Fatigue Hepatitis Hypophysitis Pneumonitis Decreased neutrophil count Neutropenia	14% of pembrolizumab group showed grade 3–5 adverse effects. 35% of paclitaxel group showed grade 3–5 adverse effects.	[30]
8	Ipilimumab BSC (best supportive care) treated	Diarrhea Fatigue Rash Colitis	23% of ipilimumab group showed grade 3–4 adverse effects. 9% of BSC group treated showed adverse effects.	[32]
9	Durvalumab plus tremelimumab (Arm A) Durvalumab monotherapy (Arm B) Tremelimumab monotherapy (Arm C) Third-line patients received durvalumab plus tremelimumab (Arm D) Second- and third-line patients receiving the combination (Arm E)	Diarrhea Fatigue Decreased appetite Colitis Pruritus Rash ALT increased	Arm A—17% showed adverse effects. Arm B—4% showed adverse effects. Arm C—42% showed adverse effects. Arm D—16% showed adverse effects. Arm E—11% showed adverse effects.	[33]

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
