# Peer review of "Therapeutic Immunomodulation in Gastric Cancer"

_cancers, 2024, doi:10.3390/cancers16030560_

Round 1

Reviewer 1 Report

Comments and Suggestions for Authors

This is a well-written review in which the authors have discussed the recent advancements in immunotherapy for advanced gastric cancer. They have summarized studies including immune checkpoint inhibitors, cancer vaccines, vascular endothelial growth factor-A inhibitors, and chimeric antigen receptor-T cell therapy that could be employed to improve the effectiveness and duration of response. The interesting feature of this review is the compilation of table summarizing the clinical trials of immune checkpoint inhibitors. However, there are minor concerns that need to be addressed to further improve the quality of the review article.

  1. The section on tumor microenvironment needs elaboration. There are studies showing the role of distant T-cells subsets in the immune microenvironment of gastric cancer (PMIDs: 32377339 and 32387455). The authors need to discuss these in their review article.
  2. It will add more strength if the authors discuss the toxicity profile and safety of immune check point inhibitors in gastric cancer as a separate subheading.
  3. The authors should consider adding a paragraph to discuss the challenges and potential strategies in immunotherapy against gastric cancers.
Comments on the Quality of English Language

None

Reviewer 2 Report

Comments and Suggestions for Authors

       In this research, the authors researched the “Therapeutic Immunomodulation in Gastric Cancer”. In my opinion, the current stage of this paper could meet the requirements of Cancers after minor revisions.

My comments are as details:

1.      The clinical transformation barrier and prospect of the therapeutic immunomodulation in gastric cancer could be discussed and predicted. The conclusion part was too plain. An in-depth outlook or conclusion should be added.

2.      In Line 122-136, the function of PD-L1 in tumor therapy sensitivity including but but only immune suppression via PD-L1/PD-1 axis. The recent new function of PD-L1 in accelerating DNA damage repair process of tumor therapies could be clearly discussed and introduced here. Some references could be added to this part including 10.1002/advs.202207608.

3.      Some minor mistakes exist in this paper. The authors should carefully check it.

4.      How the combination of therapeutic Immunomodulation in Gastric Cancer and some other solid tumors work such as combination radiotherapy, chemotherapy, and ACT or some others with PD-L1 inhibitor, Cox-2, TGF-β…? The authors should more clearly add this. Some references could be added to this part including 10.1016/j.ijbiomac.2023.127911.

5.      More tables or figures could be added to summary the therapeutic immunomodulation in gastric cancer.

Reviewer 3 Report

Comments and Suggestions for Authors

The AA did a very extensive and complete review of the potential use of immunotherapy in gastric cancer. I think that, although some studies do not show differences in terms of response in regard of tumor sub type, other studies clearly favor the use of these therapies in tumors following the mutator phenotype. I think that this issue should be emphasized as well as some related aspects: 1) there a specific tumor sub type Gastric Cancer with Lymphoid Stroma in which these therapies should be more effective 2) Are these sub types promptly identified in endoscopic biopsies or only in surgical specimens?

Also the AA should discuss whether patients with locally advanced tumors (>T1b >N0) should do these therapies peri-operatively like recommended for FLOT regimens in ESMO guidelines ?  
